# Information loss and bias in likert survey responses

**J. Christopher Westland** *

University of Illinois, Illinois, Chicago

* westland@uic.edu

**Data Availability Statement:** Data was downloaded on June 6, 2021 from https://www.kaggle.com/sjleshrac/airlines-customer-satisfaction.

**Funding:** The authors received no specific funding for this work.

## Abstract

Likert response surveys are widely applied in marketing, public opinion polls, epidemiological and economic disciplines. Theoretically, Likert mapping from real-world beliefs could lose significant amounts of information, as they are discrete categorical metrics. Similarly, the subjective nature of Likert-scale data capture, through questionnaires, holds the potential to inject researcher biases into the statistical analysis. Arguments and counterexamples are provided to show how this loss and bias can potentially be substantial under extreme polarization or strong beliefs held by the surveyed population, and where the survey instruments are poorly controlled. These theoretical possibilities were tested using a large survey with 14 Likert-scaled questions presented to 125,387 respondents in 442 distinct behavioral-demographic groups. Despite the potential for bias and information loss, the empirical analysis found strong support for an assumption of minimal information loss under Normal beliefs in Likert scaled surveys. Evidence from this study found that the Normal assumption is a very good fit to the majority of actual responses, the only variance from Normal being slightly platykurtic (kurtosis $\sim$ 2) which is likely due to censoring of beliefs after the lower and upper extremes of the Likert mapping. The discussion and conclusions argue that further revisions to survey protocols can assure that information loss and bias in Likert-scaled data are minimal.

## Introduction

Likert mappings were named for Rensis Likert, who developed them in his PhD thesis, and promoted them for the remainder of his long career [1–3]. They are mappings from responses to simple statements that are claimed to capture unobservable beliefs in a discrete scale that expresses both the direction and strength of preferences (e.g., traits, habits, consumption patterns, political orientation, and so forth). Likert scaled survey questionnaires provide the evidence for a large portion of social, corporate and government policies that are guided by surveys. This research investigates information loss and bias in mappings from the unobservable belief distributions of a population into the Likert-scaled responses on a survey instrument, asking the research question:

- Is the mapping of survey respondents' preferences into a Likert metric 'loss-less' in the sense that: (1) that the sample contains all the information from the respondents actual

**Competing interests:** The authors have declared that no competing interests exist.

preferences; and (2) that the mapping does not add information to the sample that is not in the respondents' actual preferences?

Bias and informativeness of Likert metrics have been the center of recent questions about census and political polling accuracy, and indeed may be confounded with other survey biases —e.g., problems obtaining representative samples in phone surveys [4], improper weighting by education [5], poll objectives [6] and peoples' resistance in answering poll questions [4]. Thanks partly to caller ID, average polling response in the US has fallen to only 6% in recent years, from more than half in the 1980s [7]. It is difficult to draw a sample of respondents who resemble their population. Increasing tribal, educational and non-reporting biases now affect marketing, census and social survey research.

Surveys are designed to elicit respondent preferences (e.g., for a car design), opinions (e.g., whether cars are harmful), behavior (e.g., whether brands affect purchasing), or facts (e.g., do you own a dog). Questionnaires are attractive as cheap and simple interrogative protocols. Within the past decade, survey questionnaires have been heavily automated, through companies such as the cloud-based software as a service *SurveyMonkey*. Automation has fostered a proliferation of questionnaires by investigators not particularly conversant in the methods and statistics of surveys. The automation and 'democratization' of surveys likely exacerbates failures in Likert mapping [8].

Metrics are basically distance functions, and in survey research, and points on a Likert-scales are assumed equidistant—i.e., distance between 1 and 2 is equidistant to 2 to 3; were it not, the survey would introduce 'response bias.' Survey responses may also be polytomous Rasch modeled as interval estimates on a continuum, and Likert points are levels of an attitude or trait—e.g. as might be used in consumer assessment, and scoring of performances by judges. A good Likert metric is symmetric about a middle category. Symmetric and equidistant Likert scales will behave like interval-level scales [9]. [10] showed that interval-level measurement was better achieved by a visual analogue scale [11–13]. [14] found additional grammatical biases, where balanced Likert metrics become unbalanced in interpretation; for instance when 'tend to agree' is not directly opposite 'tend to disagree.' Other biases are: (1) central tendency bias; (2) acquiescence bias; and (3) social desirability bias. [15, 16] suggest that cultural, presentation and subject matter idiosyncrasies also introduce bias. [16] point out that Asian survey responses tend to suffer more from central tendency bias than Western responses.

This research builds and analyzes a model of Likert mapping of respondent beliefs. Section 2 builds an information loss model and applies this to hypothetical situations that reflect bias and resolution in a particular situation. Section 3 empirically investigates the results of the normative models in section 2 by analyzing a 129,880 observations of 14 questions for 442 demographic groups. Finally, section 4 suggests 'best practices' to ameliorate loss and bias in survey research.

## Materials and methods

There exists a robust, empirically tested literature on preference ordering and expected utility in economics and psychology (see [17] for a summary). Surveys map unobservable human preferences, beliefs and opinions into sample items measured in some representative metric (e.g., Likert) which can be statistically summarized and analyzed. Researchers have established at least three axioms governing the topology of human preference orderings in beliefs:

1. the axiom of transitivity; $A \succcurlyeq B \text{ and } B \succcurlyeq C \Rightarrow A \succcurlyeq C$.

2. the axiom of completeness; $\forall A \text{ and } B \text{ we have } A \succsim B \text{ or } B \succsim A \text{ or both}$, and

3. the axiom of continuity; there are no 'jumps' in people's preferences.

Belief distributions adhering to these axioms are continuous, convex, differentiable and monotonic over each survey respondent's range (the support) over all possible beliefs and preferences. The survey instrument maps these into Likert metrics distributed in a polytomous Rasch model with $k$ bins (where $k$ is typically 5,7 or 9).

The geometry of Likert mapping reveals two inherent sources of bias and information loss: (1) binning which creates a discrete approximation of continuous beliefs, and (2) systematic censoring of extreme beliefs in the population.

Balanced, properly scaled Likert mappings will maintain the integrity of respondents beliefs, as can be shown in the following example. Let a particular survey instrument of $n$ responses fill the $i^{th}$ of $k$ Likert-scaled bins with probability $q_i$. If $X_i$ is an indicator variable for choice of the $i^{th}$ bin, then the possible Likert scaled outcomes are $X_1 \cup X_2 \cup \ldots X_{k-1} = \cup_{i=1}^{k-1} X_i$, with Fisher information:

$$I_{\cup_{i=1}^{k-1} X_i} = \sum_{i=1}^{k-1} \left( \frac{n}{q_i(1-q_i)} \right)$$

A Likert mapping of Gaussian beliefs $N(\mu, \sigma^2)$ which is perfectly balanced would result in $q_i$ such that the belief distribution is the limiting distribution, and Fisher information for $n$ responses that is $I_n = \frac{n}{\sigma^2}$. The Gaussian approximation for the Bernoulli mappings to the Likert-scale values had variance $\sigma^2 = \frac{q(1-q)}{n}$ and the Fisher information lost or inserted during the Likert mapping would be:

$$\sum_{i=1}^{k-1} \left( \frac{n}{q_i(1-q_i)} \right) - \sum_{i=1}^{k-1} \left( \frac{n}{q_i(1-q_i)} \right) = 0$$

## Measuring information loss

Information content in information theory can be thought of as an alternative way of expressing probability as 'entropy' of a discrete random variable, introduced in [18]. In the current context of continuous beliefs mapped into discrete Likert scales, we need a measure of the difference between two probability distributions that is valid for both continuous and discrete distributions, which in this research is the Jeffreys divergence from a reference distribution. Fisher information, self-information, mutual information, Shannon entropy, conditional entropy and cross entropy can all be mathematically derived from Jeffreys divergence. Jeffreys divergence of a target dataset is the information required to reconstruct the target given the source (i.e., minimum size of a patch).

Fisher information measures the amount of information that an observable random variable carries about an unknown parameter of a distribution that models that random variable. Formally, it is the variance of the score, or the expected value of the observed information. It always exists because it is based on actual measurements. It can be derived as the Hessian of the relative entropy.

It is essential to differentiate between the theoretical and observed Fisher Information matrices. The negative Hessian evaluated at the MLE corresponds to the observed Fisher information matrix evaluated at the MLE. In contrast, the inverse of the (negative) Hessian is an estimator of the asymptotic theoretical covariance matrix, and the square roots of the diagonal elements are estimators of the standard errors. The theoretical Fisher information matrix is based on the Fisher information metric theorem which proves that KL-divergence is directly related to the Fisher information metric.

Formally, let $l(\theta)$ be a log-likelihood function and theoretical Fisher information matrix $I(\theta)$ be a symmetrical $(p \times p)$ matrix containing the entries $I(\theta) = -\frac{\partial^2}{\partial \theta_i \partial \theta_j} l(\theta)$ for $1 \leq i, j \leq p$. The Hessian is defined as $H(\theta) = \frac{\partial^2}{\partial \theta_i \partial \theta_j} l(\theta)$ for $1 \leq i, j \leq p$ and is the matrix of second derivatives of the likelihood function with respect to the parameters. It follows that if you minimize the negative log-likelihood, the returned Hessian is the equivalent of the observed Fisher information matrix whereas in the case that you maximize the log-likelihood, then the negative Hessian is the observed information matrix.

The *observed Fisher information matrix* is $I(\hat{\theta}_{ML})$ the information matrix evaluated at the maximum likelihood estimates (MLE). The second derivative of the log-likelihood evaluated at the maximum likelihood estimates (MLE) is the observed Fisher information [19]. The optimization algorithms used in this research return the Hessian evaluated at the MLE. When the negative log-likelihood is minimized, the negative Hessian is returned. The estimated standard errors of the MLE are the square roots of the diagonal elements of the inverse of the observed Fisher information matrix. That is, the square roots of the diagonal elements of the inverse of the Hessian (or the negative Hessian) are the estimated standard errors. The inverse of the Fisher information matrix is an estimator of the asymptotic covariance matrix $Var(\hat{\theta}_{ML}) = [I(\hat{\theta}_{ML})]^{-1}$ and the standard errors are then the square roots of the diagonal elements of the covariance matrix.

The main reason to be concerned with singularities in computing Fisher Information has to do with the asymptotics—a singularity implies that the usual $\sqrt{(n)}(\hat{\theta} - \theta) \xrightarrow{D} N[0, I(\hat{\theta}) - 1]$ is not valid. Alternative formulations are provided in [20] and give the generalized asymptotic distributions, dependent on a parameter $s$ and its parity (odd/even), where $2s+1$ is the number of derivatives of the likelihood. [20] provides a unified theory for deriving the asymptotic distribution of the MLE and of the likelihood ratio test statistic when the information matrix has rank one less than full and the likelihood is differentiable up to a specific order. This is important since the likelihood ratio test uses the asymptotic distribution.

Kullback–Leibler divergence (KLD) is a 'divergence' metric. The use of the term 'divergence' as statistical distances are called, has varied significantly over time, with current usage established in [21]. [22] actually used 'divergence' to refer to the symmetrized divergence defined and used in [23], where [23] referred to this as 'the mean information for discrimination . . . per observation' while [24] referred to the asymmetric function as the 'directed divergence.'

To assure that there is no confusion in the ordering of distributions (i.e. the benchmark standard distributions versus the empirical distributions) this research uses the symmetric Jeffreys divergence, which is a true distance metric (i.e., the distance from $A \rightarrow B$ is the same as that from $B \rightarrow A$).

In practice it is not possible in advance of survey data collection to accurately balance Likert mappings; to do so you would need information about the outcome of the survey. When survey instruments are unbalanced, either by not centering responses or because they ignore extreme polarization, losses and biases can be significant as demonstrated in the two examples depicted in Fig 1, and developed in the following two examples. I have provided some parameter settings in these examples that were designed to illustrate potentially extreme situations that could occur in Likert-scaled data. While it is not my intention to explore a parameter space, as say one might do in optimization or machine learning, these do represent situations that we might expect to occur in survey practice. Further below, I analyze a large scale survey in the Airline industry to provide a real world benchmark for parameters that explores situations most likely to be encountered in practice.

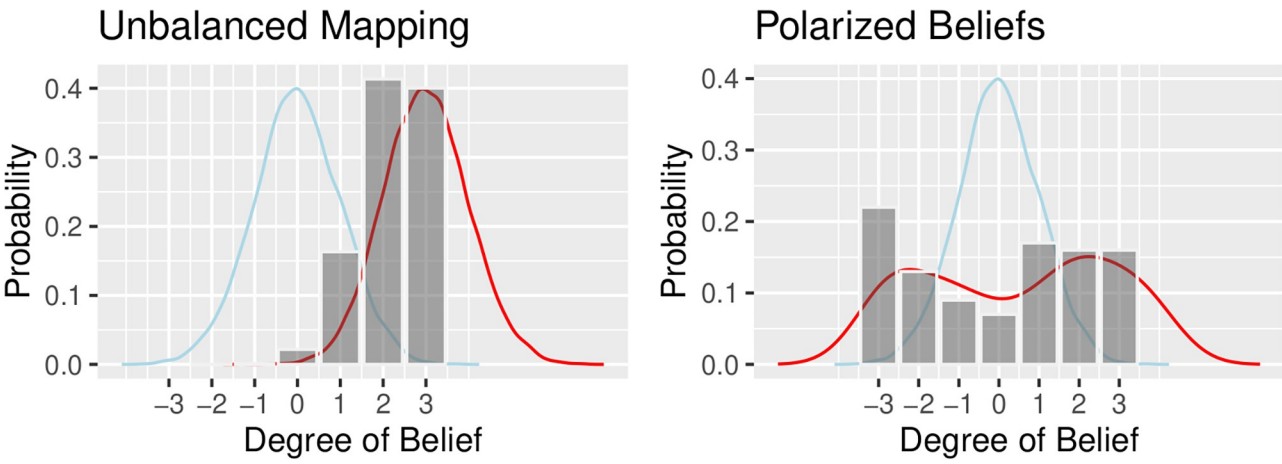

**Fig 1. Likert maps: Survey assumptions (blue) Actual beliefs (red) Likert mapping (grey).**

**Example: Unbalanced mappings with Gaussian beliefs.** A Gaussian distribution $N(\mu, \sigma)$ has Fisher information matrix:

$$I(\mu, \sigma) = \begin{pmatrix} 1/\sigma^2 & 0 \\ 0 & 2/\sigma^2 \end{pmatrix}$$

Maximum likelihood estimates of $\mu$ and $\sigma$ are *sufficient*, thus cross-correlation terms are zero and only variance (through the Cramér–Rao bound) contributes to Fisher information. Where a Likert mapping is mis-scaled so that it fails to capture extreme preference responses (e.g., see the left hand graph in Fig 1) it *censors* data that we know exists outside its range. Censoring (in contrast to truncation) remaps beliefs outside the Likert support into the rightmost or leftmost extreme value of the Likert scale. Table 1 summarizes Fisher information in Gaussian beliefs censored by a Likert mapping.

Table 1 shows that for balanced Likert mappings, information loss and bias are insignificant. Table 2 summarizes an unbalanced mapping example, the Likert scale is built assuming respondents have $N(\mu_{standard}, \sigma_{standard}) = N(0, 1)$ while the actual responses are significantly biased with distribution $N(\mu_{actual}, \sigma_{actual}) = N(3, 1)$.

Note that censoring *adds* false information; the 'gains' (negative percentages) in information result from artificial inflation of extreme right-hand value + 3 in the Likert mapping.

**Example: Beta beliefs that express polarization.** Polarized beliefs can be simulated using a $Beta(\alpha, \beta)$ distribution, where $\alpha, \beta < 1$ (e.g., as depicted in the right-hand graph in Fig 1) supported on a −3, 3 support on a 7-point Likert scale. The p.d.f. of a Beta with [*upper*, *lower*]

**Table 1. Fisher information censored by remapping $N(\mu = 0, \sigma = 1)$ beliefs.**

| | Standard Gaussian Distribution | Censored Standard Gaussian Distribution |
|---|---|---|
| $\mu \times \mu$ | 9757 | 9990 |
| $\mu \times \sigma$ | 0 | 0 |
| $\sigma \times \sigma$ | 19515 | 19979 |
| $\sigma \times \mu$ | 0 | 0 |

**Table 2. Fisher information for $N(\mu = 3, \sigma = 1)$ beliefs with an 'unbalanced' Likert mapping.**

|  | Actual Beliefs | Censoring | Binning | % Censoring Loss | % Binning Loss |
|---|---|---|---|---|---|
| $\mu \times \mu$ | 10187 | 14762 | 8444 | -45% | 43% |
| $\mu \times \sigma$ | 0 | 0 | 0 | 0% | 0% |
| $\sigma \times \sigma$ | 20374 | 29524 | 7418 | -45% | 75% |
| $\sigma \times \mu$ | 0 | 0 | 0 | 0% | 0% |

**Table 3. Fisher information for $Beta(\alpha = 0.5, \beta = 0.5)$ beliefs mapped to a Likert-scale.**

|  | Actual Beliefs | Censoring | Binning | % Censoring Loss | % Binning Loss |
|---|---|---|---|---|---|
| $\alpha \times \alpha$ | 172 | 144 | 228 | 16% | -58% |
| $\alpha \times \beta$ | 87 | 73 | 116 | 16% | -58% |
| $\beta \times \beta$ | 148 | 124 | 196 | 16% | -58% |
| $\beta \times \alpha$ | 87 | 73 | 116 | 16% | -58% |

bounds of support is

$$B(x|\alpha, \beta) = \frac{\Gamma(\alpha + \beta)}{\Gamma(\alpha)\Gamma(\beta)} \left( \frac{x + upper}{(upper - lower + 1)} \right)^{\alpha - 1} \left( 1 - \frac{x + upper}{(upper - lower + 1)} \right)^{\beta - 1}$$

Fisher information in a *Beta*-distributed random variable X is:

$$I(\alpha, \beta) = \begin{bmatrix} \text{var}[\ln X] & \text{cov}[\ln X, \ln(1 - X)] \\ \text{cov}[\ln X, \ln(1 - X)] & \text{var}[\ln(1 - X)] \end{bmatrix}$$

Table 3 summarizes Fisher information where the Likert scaling assumes $N(\mu_{standard}, \sigma_{standard}) = N(0, 1)$ while actual responses are distributed $B(\alpha_{actual}, \beta_{actual}) = B(0.5, 0.5)$.

For $X \sim Beta(\alpha, \beta)$ then $\text{Var}[\ln X] = \text{Var}[\ln(1 - X)] = \text{Cov}[\ln X, \ln(1 - X)] = \psi(\alpha) - \psi(\alpha + \beta)$ where $\psi$ is the trigamma function. This explains the symmetric scaling of the percentage errors, and once again only variance contributes to Fisher information. Because the extreme mappings are more or less balanced, the impact of censoring is less important in the *Beta* example than changes in resolution due to binning, which injects artificial information into the survey statistics.

## Results

The prior analysis postulated example situations which may or may not be commonplace, that suggest Likert responses could in practice lead to erroneous conclusions. The current section evaluates empirical results from a professionally conducted airline customer satisfaction survey to provide insight into whether the preceding problems are common in practice. The dataset used in this study was extracted from a professional 2015 survey of customer satisfaction by a major US airline, who released the data on the condition of remaining anonymous. The curated database is available in three locations on Kaggle, at https://www.kaggle.com/sjleshrac/airlines-customer-satisfaction, www.kaggle.com/johndddddd/customer-satisfaction and at www.kaggle.com/teejmahal20/airline-passenger-satisfaction and was downloaded on June 6, 2021 from the first source on this list. The dataset has been the subject of a number of machine learning and sentiment analysis studies documented on the Kaggle site.

**Table 4. Demographic groups in this research defined by factors and levels.**

| Factor | Levels | Level 1 | Level 2 | Level 3 |
|---|---|---|---|---|
| satisfaction | 2 | Dissatisfied | Satisfied | ~ |
| gender | 2 | Female | Male | ~ |
| customer.type | 2 | Disloyal | Loyal | ~ |
| travel.type | 2 | Business | Personal | ~ |
| travel.class | 3 | Business | Economy | Economy+ |
| age | 3 | <21 | 21-60 | >60 |
| distance | 3 | <100 | 100-1000 | >1000 |
| departure.delay | 3 | ontime | <1hr | >1hr |
| arrival.delay | 3 | ontime | <1hr | >1hr |

I have chosen a consumer sentiment dataset that is extensive, and has been well researched over the past six years. One of the most common commercial applications of Likert-scaled surveys is the assessment of consumer sentiment, so it was my impression that such a dataset would be most germane to the particular research problems I was addressing. A large scale survey such as this, prepared by a professional consulting firm, will be better controlled and curated than would be typical for the average social science Likert-scaled dataset. The acquisition of this dataset was performed under appropriate controls, and similar studies would be replicable. The sample sizes were very large by academic standards, and encompassed a rich set of demographic-behavioral factors and customer satisfaction factors.

This is a US airline, and results are all from US passengers. The survey is huge by academic research database standards. The respondents were drawn from 442 different behavioral-demographic groups summarized by existing permutations of the factor levels in Table 4. The results are extremely robust, as shown in the reported statistics. The factors and levels in Table 4 were those for which the airline had internal seat occupancy and upgrade algorithms, largely focused on determining seat prices. The airline actively manages these particular factors and factor levels and considers them to be the key success factors controlled by the airline's profitability. The survey was conducted by a consulting firm versed in survey research, and was controlled at a professional and high technical standard. The survey technical standards exceed that for most academic research surveys. The R code used to create this research paper is uploaded to Kaggle. The survey is huge by research database sizes, consisting of 129,880 independent responses to 14 Likert-scaled question responses (Table 5) from each customer concerning satisfaction with a particular factor managed at the airline. The respondents were drawn from 442 different behavioral-demographic groups summarized by existing permutations of the factor levels in Table 4. The results are extremely robust, as shown in the reported statistics. The specific factors that were Likert-scale surveyed were those for which the airline had internal programs and staff dedicated to assuring that customers were well served on their airline. They are paying money to manage these factors, and consider them to be the key success factors controlled by the airline's management

There are $2^4 \times 3^5 = 3888$ unique combinations of factor values (Table 4) implying 3888 potential behavioral-demographic subgroups of passengers who would be expected to respond to the questionnaire each in their own unique way. Only 1375 different subgroups existed in the actual dataset, and only 442 of these subgroups had more than 20 observations, which was considered the minimum acceptable for fitting to a Normal distribution or for Central Limit Theorem convergence. These 442 subgroups with different response behaviors and biases were studied in this research. This reduced the total number of responses in the survey from

**Table 5. Summary statistics for raw data.**

| Factor | Mean | Std.Dev | Skewness | Kurtosis |
|---|---|---|---|---|
| Seat.comfort | 2.839 | 1.393 | -0.09186 | 2.057 |
| Departure.Arrival.time.convenient | 2.991 | 1.527 | -0.25228 | 1.911 |
| Food.and.drink | 2.852 | 1.444 | -0.11681 | 2.013 |
| Gate.location | 2.990 | 1.306 | -0.05306 | 1.910 |
| Inflight.wifi.service | 3.249 | 1.319 | -0.19112 | 1.879 |
| Inflight.entertainment | 3.383 | 1.346 | -0.60482 | 2.467 |
| Online.support | 3.520 | 1.307 | -0.57536 | 2.189 |
| Ease.of.Online.booking | 3.472 | 1.306 | -0.49171 | 2.089 |
| On.board.service | 3.465 | 1.271 | -0.50526 | 2.215 |
| Leg.room.service | 3.486 | 1.292 | -0.49643 | 2.159 |
| Baggage.handling | 3.696 | 1.156 | -0.74303 | 2.762 |
| Checkin.service | 3.341 | 1.261 | -0.39244 | 2.206 |
| Cleanliness | 3.706 | 1.152 | -0.75599 | 2.791 |
| Online.boarding | 3.353 | 1.299 | -0.36649 | 2.062 |

129,880 to 125,387 (i.e., reduced by 4493 responses) and 442 distinct behavioral-demographic groups.

An exploratory comparison of the empirical distribution of Likert-scaled data from the Airline Customer Satisfaction dataset used below, each to a *Normal*(3, 1), *Poisson*(3) and *Beta*(.5, .5) random variables, showed only insignificant difference between Jeffreys divergence and KLD for this dataset.

Table 5 summarizes the first four moments of Likert responses for the entire 129,880 responses across the dataset. Note that the assumptions made in the normative analysis of Likert-scaled metric information content are generally adhered to—mean of 3 (center of 5-point Likert scale), standard deviation of $\sim 1$, skewness $\sim 0$ (centered responses), with the responses being slightly platykurtic due to the truncation of beliefs above Likert-5 and below Likert-1.

Figs 2 through 5 show that our standard assumption of 5-point Likert responses accurately reflecting a *Normal*($\mu = 3$, $sd = 1$) with 1-point on the Likert scale being equal to the $\sigma$ standard deviation of the Normal assumption. In addition, Skewness is zero, and kurtosis is consistently platykurtic.

## Empirical results: The information penalty of an incorrect belief assumption

The Fisher information metric theorem proves that KLD (and thus Jeffreys divergence) is directly related to the Fisher information metric, with the Fisher Information Matrix being the Hessian of the KLD between two distributions evaluated at the MLE. I use Jeffreys divergence in the empirical analysis for this paper to measure the information penalty (distance) between actual Likert responses, and an assumed distribution of respondents' beliefs. Jeffreys divergence provides a measurement of how far the distribution $Q$ is from the distribution $P$. Jeffreys divergence is used in areas such as clutter homogeneity analysis in radar processing, and KLD is used in ruin theory in insurance and in computing Bayesian information gain in moving from a prior distribution to a posterior distribution [22, 25].

Jeffreys divergence is employed here for succinctness and clarity—whereas Fisher information is a matrix whose dimension depends on the number of parameters of the distribution,

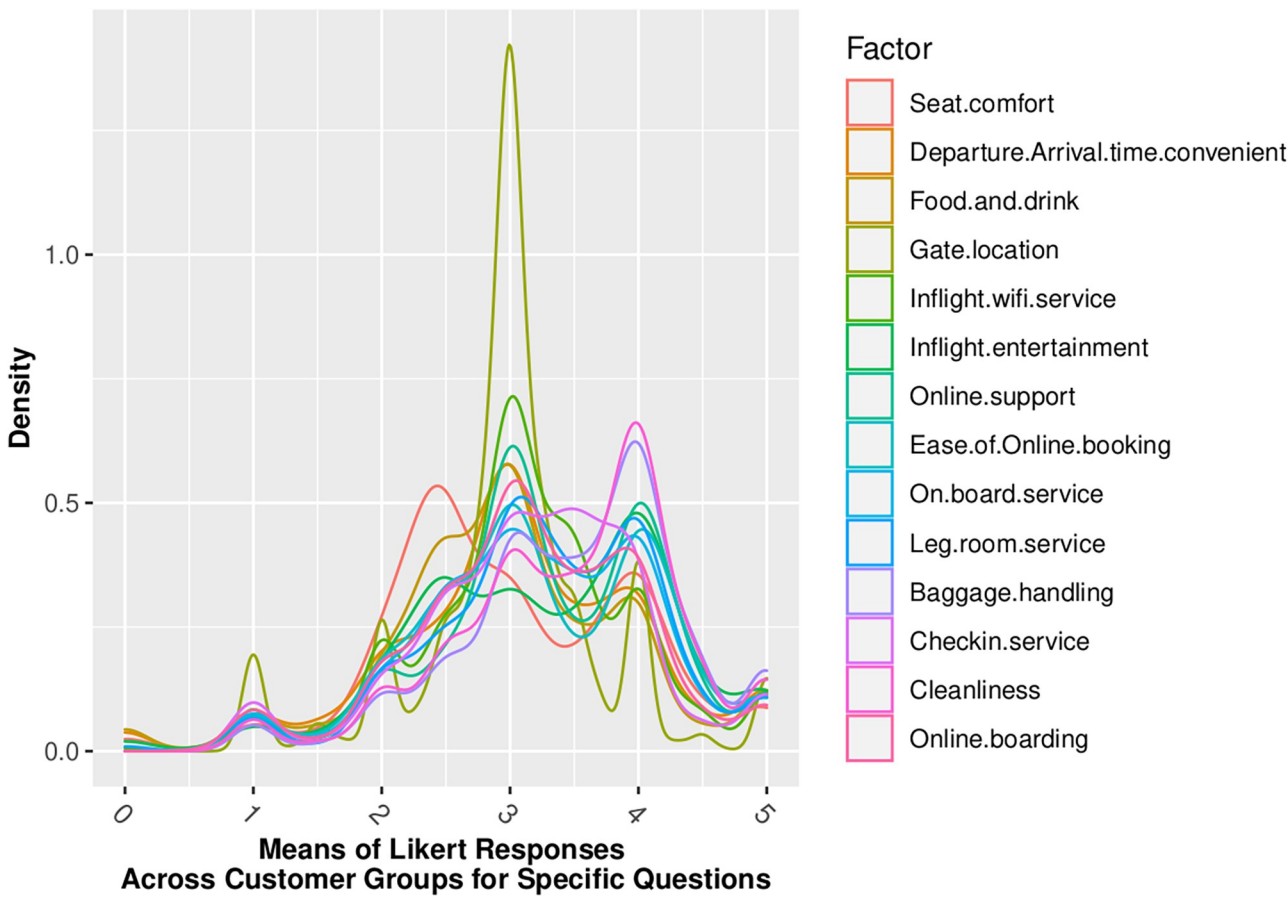

**Fig 2. Means of likert responses by demographic.**

Jeffreys divergence is a single distance metric. Jeffreys divergence reports the amount of information lost because of particular modeling assumptions—in the current case, the assumptions about the underlying respondent beliefs that are being mapped into a Likert-scaled metric.

Figs 6 through 8 summarize my analysis, revisiting the questions of information loss which I analyzed in normative models in the first part of this paper. Figs 6 through 8 present density graphs of the Jeffreys divergence 'information penalty' across all of the 442 behavioral-demographic groups, for each of the 14 customer satisfaction factors, under 3 belief assumptions—Normal, Poisson and Beta distributed beliefs.

Across the 14 factors for which we have Likert-scaled responses, the assumption of Normal belief suffers the least information loss, as summarized in Table 6. This is consistent with insights gained from the review of distributional statistics of the full dataset responses in Figs 2 through 5, which support the validity of prior assumptions of Normal beliefs in well-controlled Likert-scaled survey responses.

## Discussion and conclusions

Polarized survey responses are common in research on political and social issues in North America, and thus the problem addressed in this research is an important one. Survey bias towards boundary-inflated responses among polled Americans, and midpoint-inflated responses among Asians have been repeatedly documented and called out as a challenge to

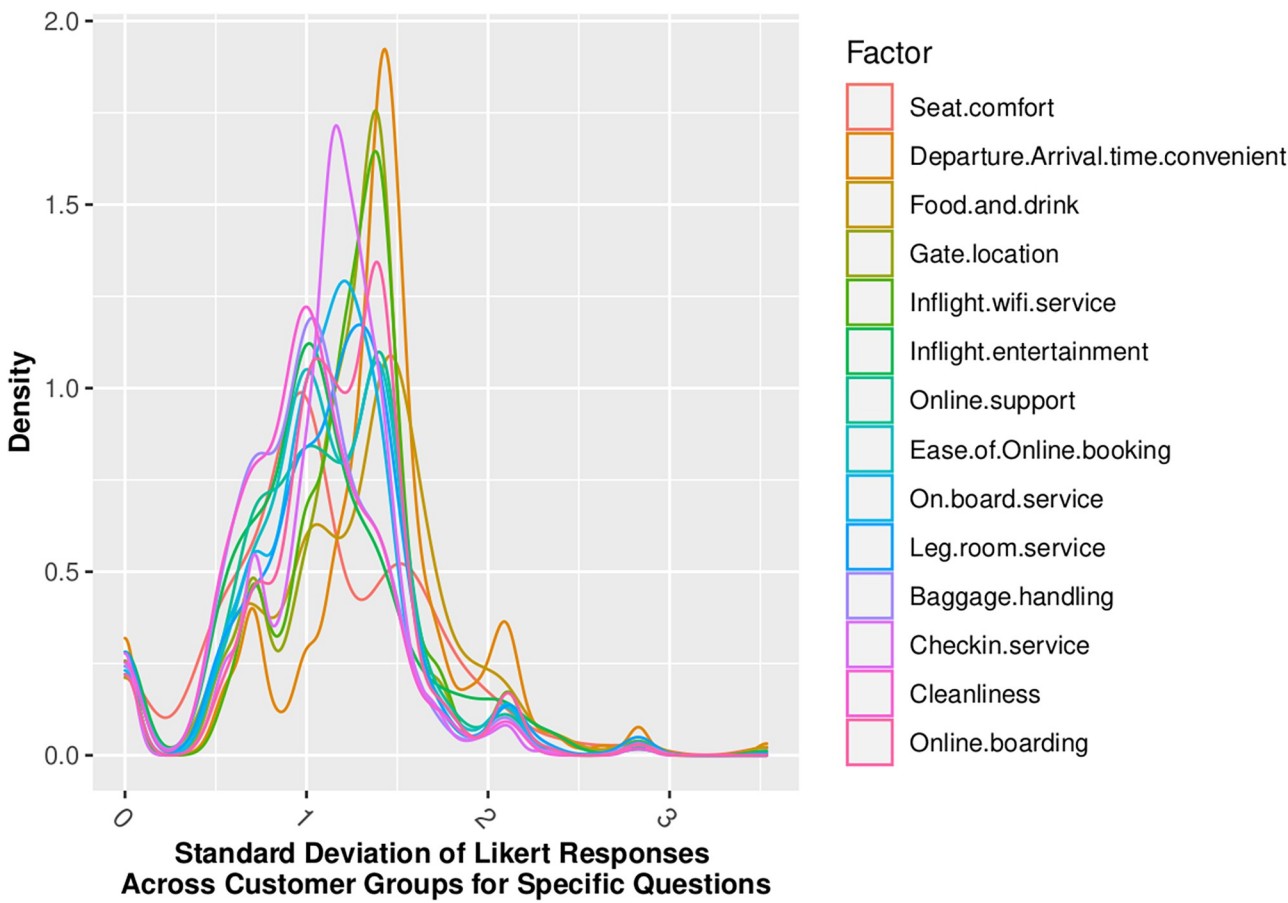

**Fig 3. Standard deviation of likert responses by demographic.**

survey based research (e.g., see [16, 26, 27]) with the Pew Research Center [28] describing such bias as a major challenge to democracy and a consistent problem in their surveys.

Cross-sensory response research in [29], specifically studies in the human taste response to music, has pioneered Bayesian alternatives to frequentist analysis of Likert-scaled data. In [29] a sample of 1611 participants tasted one sample of chocolate while listening to a song that evoked a specific combination of cross-modal and emotional consequences. The researchers addressed difficulties in interpreting frequentist statistical tests using discrete, categorical responses by applying a Bayesian model to quantify the information content of a response. The approach used in [29] is well suited to sentiment analysis problems that have long been analyzed using structural equation models and frequentest Neyman-Pearson hypothesis tests [30, 31].

Data collected for [29] study showed strong non-symmetric behavior among the bounded scales, with large numbers of respondents selecting extreme values close to the boundaries, which contradicted the assumptions of traditional multivariate regression approaches to analysis, because residuals could not be Gaussian distributed were responses at the boundaries of the response space. These sorts of polarized responses are quite common in research on political and social issues in North America, and thus the problem addressed is an important one. Often they are modeled as zero-inflated Gaussian or Poisson distributions, though too often, no accommodations are made at all for the analysis for the zero-inflated data.

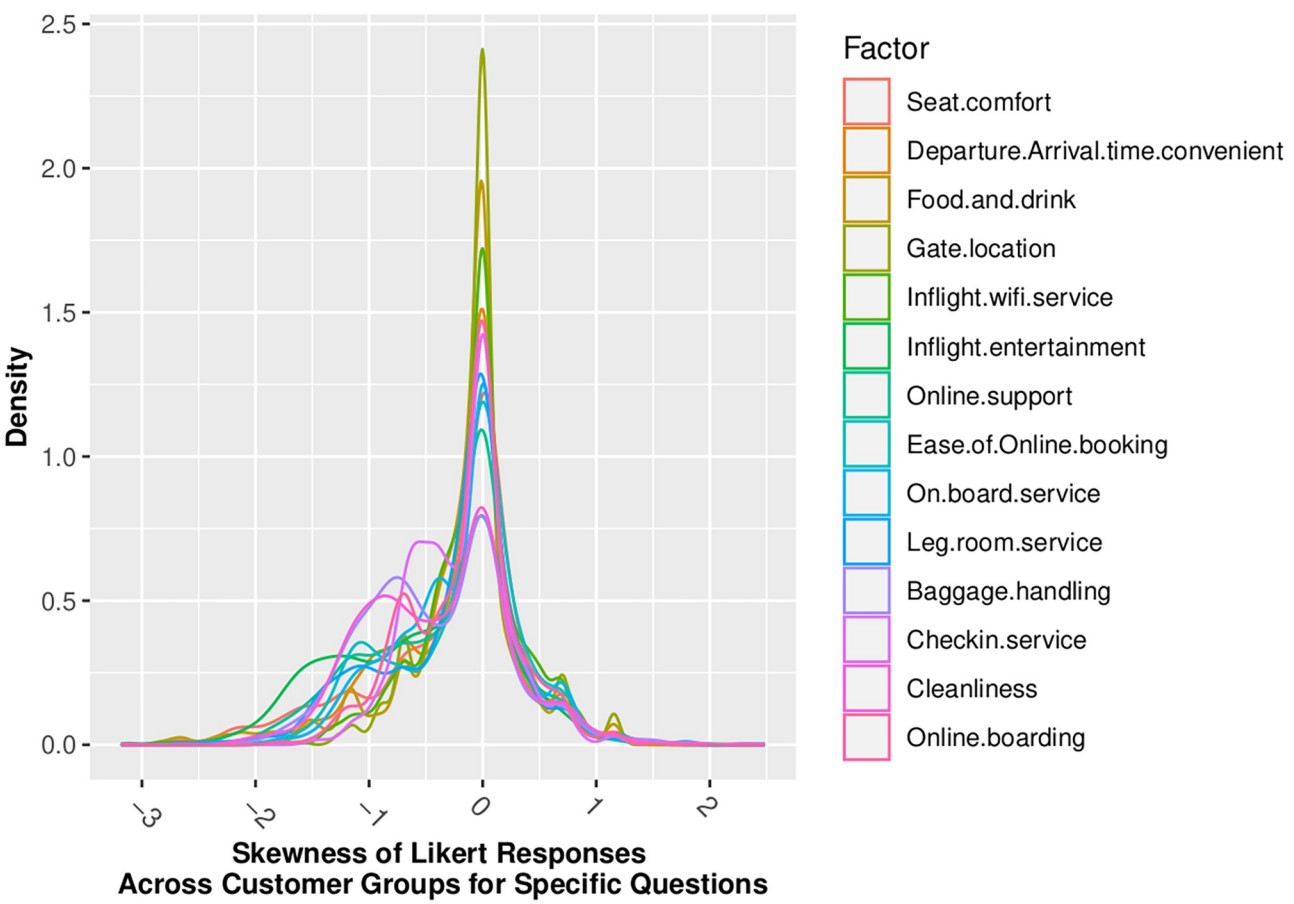

**Fig 4. Skewness of likert responses by demographic.**

In order to overcome this problem [29] remapped each outcome *j* for each individual *i* into a unit (0, 1) range. They then used Bayesian, multi-response, multivariate, logit-normal distribution with outcome-specific intercepts and slopes, and common covariance structure across outcome measures, following the methodology in [32]. The logit-normal distribution can take a variety of shapes, e.g., U-shapes and J-shapes. More importantly, they are designed to specifically address the zero-inflated data distributions that arise in particularly polarized survey responses.

A Bayesian multi-response version of the multivariate logit-normal regression model was used in [29]. Outcome-specific intercepts and slopes were needed since the association of each co-variate with each of the responses could significantly differ. They also take advantage of the inherent high-correlation of responses due to individual consistency in responses, and to social and cultural clustering of beliefs (and responses) in survey data, through joint modeling of all the outcomes, allowing the borrowing of information between responses. The Bayesian multi-response version of the multivariate logit-normal regression model presented in [29] provides a flexible, scalable and adaptive model where reliance on the central limit theorem can be questionable. Additionally, where available, they provide a natural form to incorporate any prior information either from prior studies or from expert opinion. The transformations specified in [29] result in a model error term (representing features not captured by the data) which is multi-Normal, allowing for calculation with available statistical software.

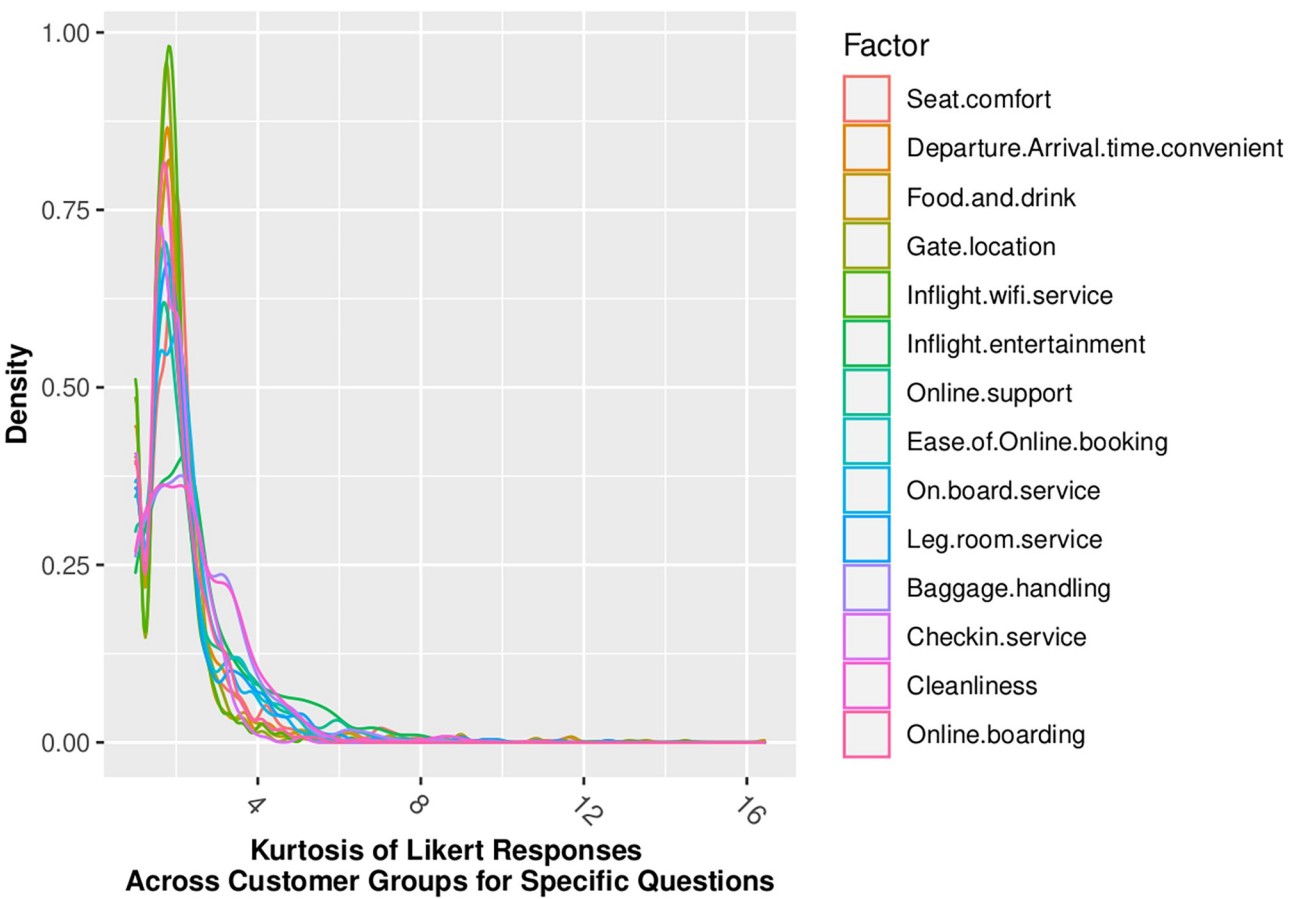

**Fig 5. Kurtosis of likert responses by demographic.**

## Suggestions for 'Best Practice' in survey use of likert metrics

This research suggested several examples to show that in theory, Likert mappings could be lossy and biased under a specific set of circumstances and using a balanced, centered and specific design for each of the individual questions on the survey instrument. Furthermore, I provided examples of situations in which:

1. the variance of the sample standard deviation from a single Likert-scale point will result in an increasing information loss;

2. the location of the belief distribution mean fails to coincides with the central Likert bin, and the survey instrument is 'unbalanced';

3. the Likert mapping depends on the mean of the belief distribution and is sensitive to the Likert metric being 'balanced';

4. where the respondents' opinions are extremely polarized with respondents choosing to extremely agree or extremely disagree with the assertion, or in contrast where respondents demur and choose the center of the Likert scale;

5. where where the respondents' opinions reflect universally held strong beliefs, either negative or positive, and;

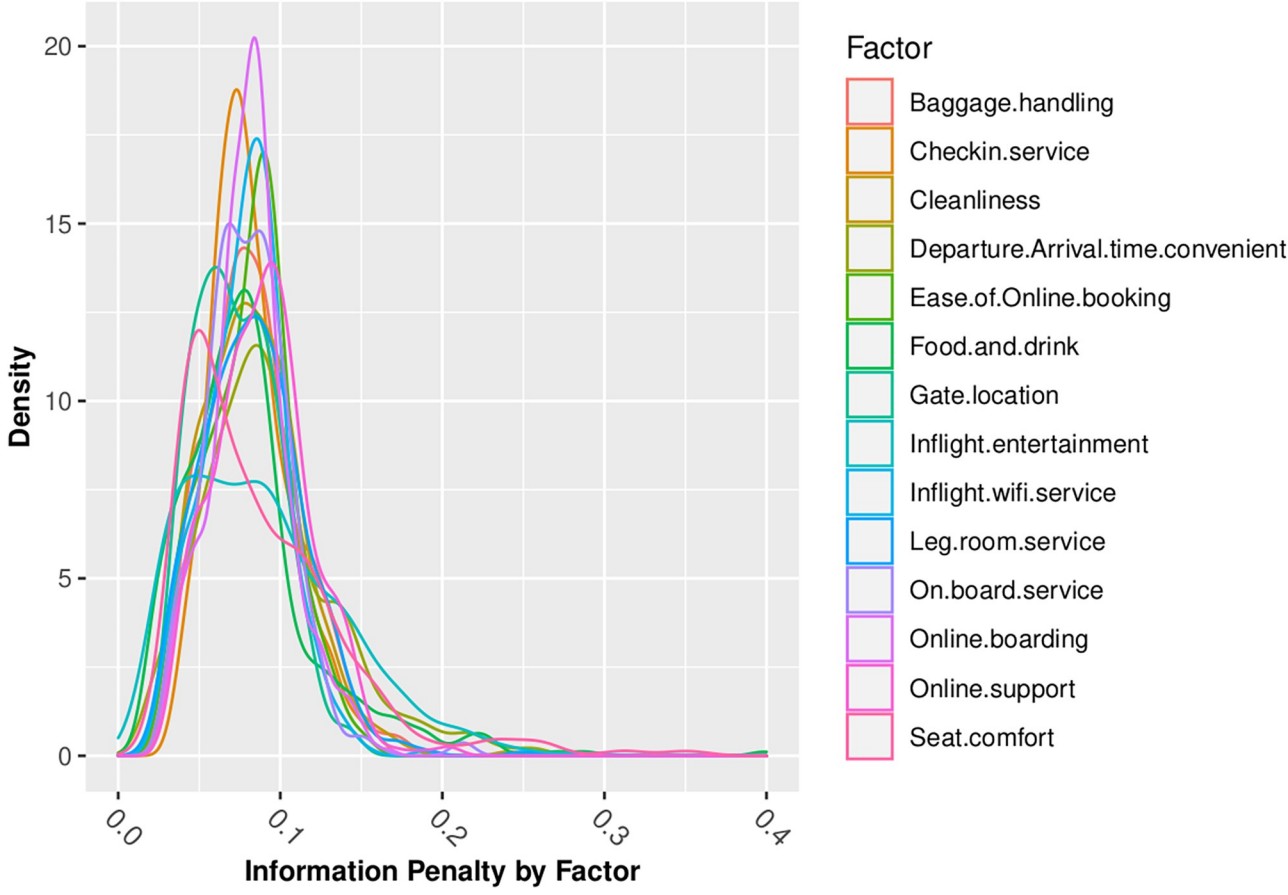

**Fig 6. Information penalty (Jeffreys Divergence) assuming normal beliefs.**

6. where censoring and binning both add information that was not in the original data.

Though the examples suggest many ways in which Likert metrics can lead to incorrect conclusions, my empirical results suggest that these problems do not assert themselves in most typical survey studies, with commonly encountered respondent behavioral-demographic profiles.

The study analyzed involved airline passengers with a broad range of demographics and behaviors. The differences in responses was small across all of the demographic groups, as reflected in the Jeffreys divergence information penalty for any particular *a priori* assumption about respondent beliefs. An assumption of Normal prior beliefs, with the mean of the belief distribution corresponding to the midpoint in the Likert scale, produced minimal information losses. The empirical analysis strongly supports current protocols and assumptions in the conduct of Likert scaled survey research.

This is reassuring, but nonetheless, it is difficult to know enough about population beliefs to design *a priori* a perfectly balanced, centered and specific survey instrument without having already completed at least a limited survey and data analysis. In practice, there exist survey protocols in which evidence-based interactive design of surveys has successfully resolved or ameliorated problems in instrument design, though these are typically ignored in most *Survey-Monkey* style implementations. Interactive, multi-step methods can involve: (1) pretesting, (2) invoking optimal stopping based on an objective function, or (3) implementing redundancy in

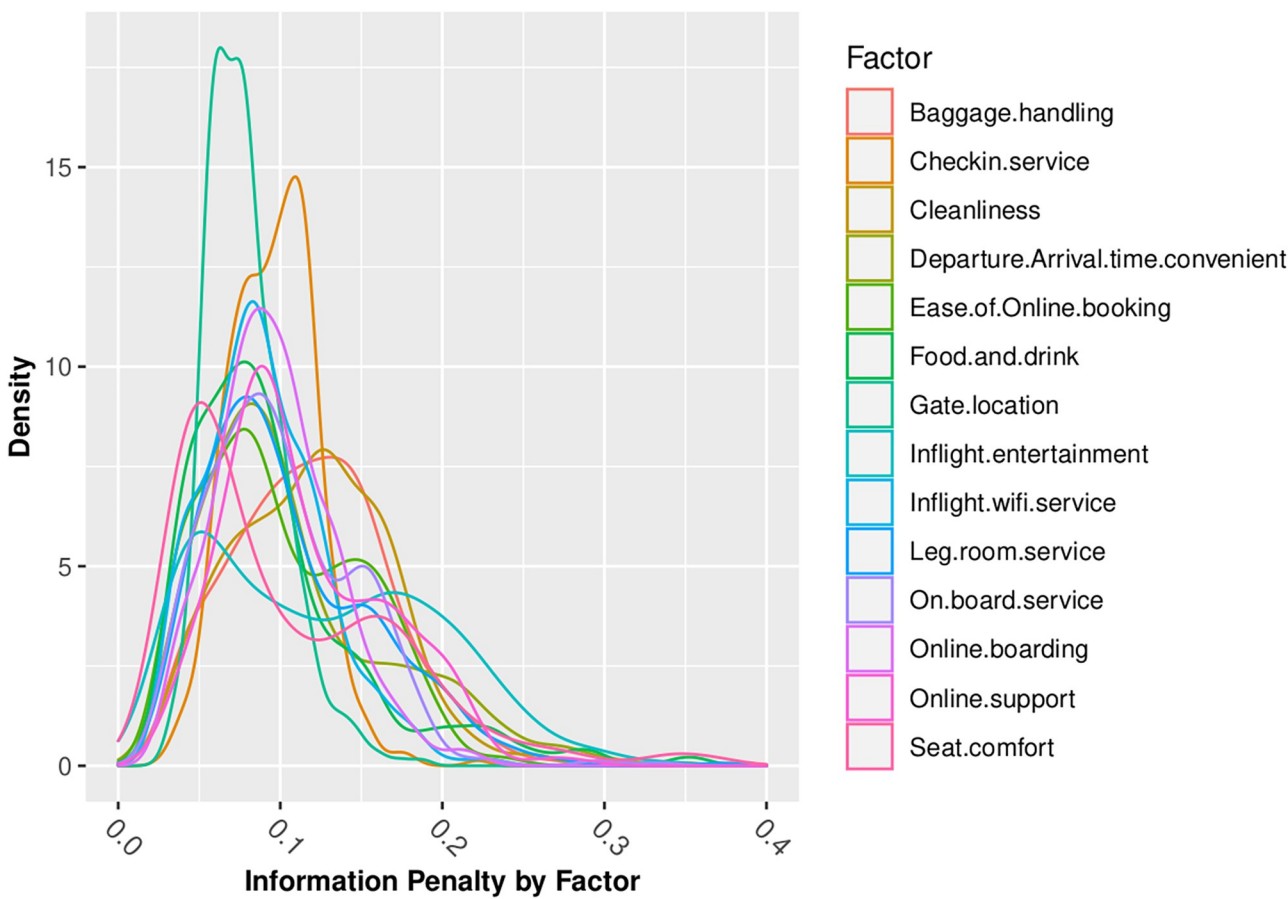

**Fig 7. Information penalty (Jeffreys Divergence) assuming poisson beliefs.**

sampling, along with general sentiment assessment to scale and center responses during the survey execution.

Finally, there is much to learn from reviewing survey strategies and analysis of responses that have been successfully employed in two research areas—polygraph protocols and clinical trials in medicine. These can potentially provide 'best practice' guidance for Likert-scaled survey research design. The most rigorous survey protocols appear in polygraph testing, partly because these protocols have access to enormous amounts of emotional responses (unavailable in *SurveyMonkey* type surveys) in addition to a subject's verbal responses. Polygraph protocols are obsessive about balancing questions, centering the response scale, and assuring that interrogation spans the gamut of the belief distribution support [33]. This happens interactively during the interview process, and polygraph interrogators are constantly adjusting and rechecking responses to questions as the interview progresses. The same question will be asked repeatedly to assure that respondents' answers are consistent and honest. In addition, polygraph interrogators ask a variety of questions besides the primary relevant statements that provide information supporting the research objective; these are used in 'fine tuning' the survey instrument. They include: (1) *irrelevant* statements designed to identify subjects 'gaming' the survey, *truthfulness* statements describing behavior that the majority of subjects have been involved in, to detect habitual liars, and *sacrificial* statements designed to absorb the initial response to a relevant issue and to set the context so that subsequent statements elicit consistent responses [33].

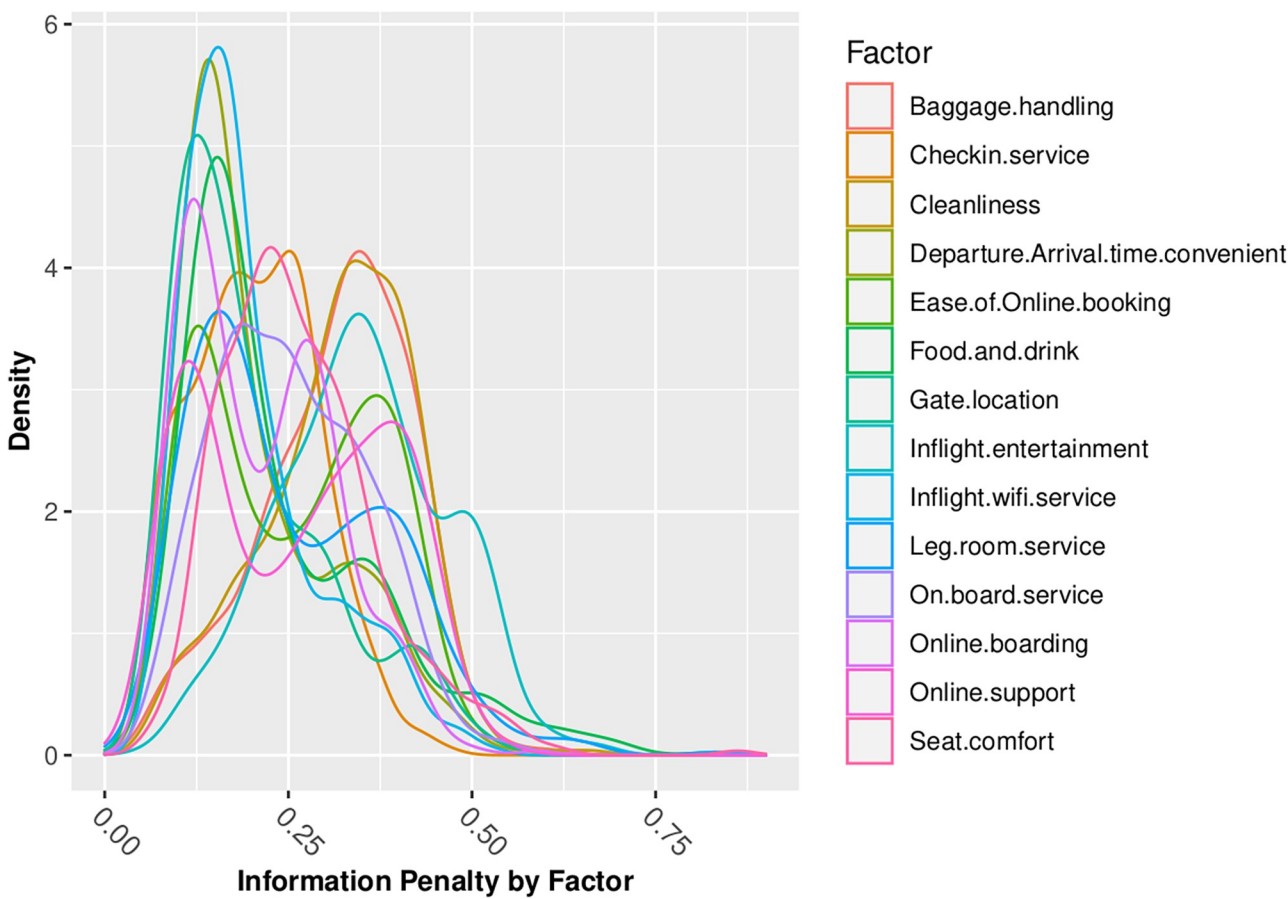

**Fig 8. Information penalty (Jeffreys Divergence) assuming beta beliefs.**

Polygraph protocols yield smaller type I and type II errors than questionnaires that lack the controls provided by sacrificial, irrelevant and truthfulness statements to benchmark the survey subject's mood, cooperativeness and seriousness about the survey [34]. In criminal law, 'Blackstone's ratio' suggests that 'It is better that ten guilty persons escape than that one innocent suffer' [35] inherently biasing judgments towards minimizing false negatives. This is why polygraphs are inadmissible in court, even though they outperform other survey protocols.

Clinical trials in medicine and pharmaceuticals have employed stopping rules, particularly in Phase I clinical trials [36]. The simplest stopping rules target a given precision for minimum cost of testing. Researchers will also review whether successive samples provide evidence that the parameter of interest is changing, by: (1) examining patterns of observed responses, and (2) using missing data methods to impute missing responses [37]. Optimal stopping models are widely used in machine maintenance, economics, and finance. As in clinical trials, Likert

**Table 6. Statistics of Jeffreys divergence information penalty for all demographic groups.**

| Distribution | Mean | Std.Dev | Skewness | Kurtosis |
|---|---|---|---|---|
| Normal | 0.0835 | 0.0365 | 3.749 | 67.11 |
| Poisson | 0.1047 | 0.0535 | 2.217 | 24.97 |
| Beta | 0.2536 | 0.1208 | 0.764 | 5.07 |

scaled survey instruments could be optimally designed to collect initial responses, where these would be tracked and analyzed, then revised to create balanced, centered and specific questions for the next round of sampling; applied again, responses collected and analyzed, revised again; and so forth until loss and bias are within a critical range.

## Author Contributions

**Conceptualization:** J. Christopher Westland.

**Data curation:** J. Christopher Westland.

**Formal analysis:** J. Christopher Westland.

**Investigation:** J. Christopher Westland.

**Methodology:** J. Christopher Westland.

**Project administration:** J. Christopher Westland.

**Resources:** J. Christopher Westland.

**Software:** J. Christopher Westland.

**Supervision:** J. Christopher Westland.

**Validation:** J. Christopher Westland.

**Visualization:** J. Christopher Westland.

**Writing – original draft:** J. Christopher Westland.

**Writing – review & editing:** J. Christopher Westland.

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
