## [Decision Letter · Decision Letter 0]

23 Feb 2022

PONE-D-21-19317Information Loss and Bias in Likert Survey ResponsesPLOS ONE

Dear Dr. Westland,

Thank you for submitting your manuscript to PLOS ONE. After careful consideration, we feel that it has merit but does not fully meet PLOS ONE’s publication criteria as it currently stands. Therefore, we invite you to submit a revised version of the manuscript that addresses the points raised during the review process.

I want to apologize for the unusually long time that this review took to be completed. There were delays with the second reviewer and I wanted to make sure that your paper was read by true experts. I hope you will find their comments useful. Overall, we all agree that this is a very informative paper. In addition to answer the reviewers suggestions, I want you to make an extra effort to make the implications of your results and your recommendations very clear to a wider audience. We look forward to reading the revised version. 

We look forward to receiving your revised manuscript.

Kind regards,

Carlos Andres Trujillo, PhD

Academic Editor

PLOS ONE

Journal Requirements:

3. Thank you for stating the following financial disclosure: "The funders had no role in study design, data collection and analysis, decision to publish, or preparation of the manuscript."

Reviewers' comments:

Reviewer's Responses to Questions

**Comments to the Author**

1. Is the manuscript technically sound, and do the data support the conclusions?

Reviewer #1: Yes

Reviewer #2: Partly

2. Has the statistical analysis been performed appropriately and rigorously? 

Reviewer #1: Yes

Reviewer #2: Yes

3. Have the authors made all data underlying the findings in their manuscript fully available?

Reviewer #1: Yes

Reviewer #2: Yes

4. Is the manuscript presented in an intelligible fashion and written in standard English?

Reviewer #1: Yes

Reviewer #2: Yes

5. Review Comments to the Author

Reviewer #1: The author showed an interesting problem together with rigurous mathematical arguments. The manuscript also has an interesting application. I would appreciate if you could read my comments in the attached pdf.

Reviewer #2: To measure the closeness between the two probability distributions, the Kullback Leibler divergence

The paper focuses on minimizing the bias and informativeness of Likert metrics distribution using the Kullback-Leibler divergence, KLD, from the actual responses empirical distribution and an assumed distribution of respondent beliefs through the information penalty.

KLD is a divergence measure, not a distance metric. Thus, the order of the two distributions matters. KLD(Q, P) is not equal to KLD(P, Q). The use of divergence and distance is confusing. Moreover, in the setup, it is unclear how the order of the distributions is.

The assumed distributions used in the paper are standard for the beliefs. However, in real applications, the implementation of much more complex distributions is appropriate and natural. For example, a mixture of distributions, zero-inflated distributions, among others.

Under the complexity issue, the Hessian matrix is singular, which leads to the impossibility of measuring the information penalty. Therefore, in this situation, the solution requires the application of different methods.

It is unclear how is designed the strategy to set optimally the hyper-parameters of the ideal belief distribution.

The empirical application is outstanding and informative, from which the minimal information loss found is under Normal beliefs in Likert scaled surveys.

6. PLOS authors have the option to publish the peer review history of their article (what does this mean?). If published, this will include your full peer review and any attached files.

Reviewer #1: No

Reviewer #2: **Yes: **hector zarate

---

## [Author Response · Author response to Decision Letter 0]

14 May 2022

Author Response to Reviewer Comments on PONE-D-21-19317 Information Loss and Bias in Likert Survey Responses

J. Christopher Westland

2022-04-23

Reviewer’s Assessment of the Suitibility of Research

Comment Reviewer #1 Response Reviewer #2 Response

1. Is the manuscript technically sound, and do the data support the conclusions? Yes Partly (I have responded below)

2. Has the statistical analysis been performed appropriately and rigorously? Yes Yes

3. Have the authors made all data underlying the findings in their manuscript fully available? Yes Yes

4. Is the manuscript presented in an intelligible fashion and written in standard English? Yes Yes

Reviewer Comments to the Author

Reviewer #1:

The author showed an interesting problem together with rigorous mathematical arguments. 

The manuscript also has an interesting application. 

Thank you. I appreciate the compliment and hopefully have addressed remaining questions concerning the methods, results and writing.

Main points

The paper sets forth the task of examining distributional implications of Lik-

ert response surveys. The latter are widely used accross many social disciplines

such as marketing, opinion polls, economic disciplines and health surveys among

many others. Being discrete categorical metrics, they can not only lose signifi-

cant information from the real-world mappings beliefs, but also generate signif-

icant biases into the statistical analysis. The manuscript sets forth the task of

showing through probabilistic arguments when do Likert scales do what they are

supposed to do and when things can go wrong (strong polarization and beliefs).

Such problems practically disappear when beliefs are Normal beliefs in Likert

scaled surveys. The Normal fit works pretty well under many circumstances.

The manuscript recommends using Likert-scaled surveys to allow minimal bias

and information loss

Author Response

I appreciate that the reviewer has accurately and succinctly synopsized the main results of my research.

Recommendation

I think the topic is very important for many scientific studies that researchers

know about how to work better using Normal Likert scaled surveys. I think it

would be useful to include some recent literature that have takcled to model

customer preferences from a different perspective to avoid the shortcomings of

loss of information and biases using Likert scales. An example of paper is one

on an alternative Bayesian modelling of Likert distributions such as Reinoso-

Carvalho et al. (2020). They use discrete scale that in practice in contious by

the use of a Bayesian Logit-Normal distribution. In this sense, there has been an

interesting discussion from the Bayesian perspective and literature dealing with

such problems posed by the Likert scales and it would be interesting that the

author fills that discussion in the introduction and motivation of the manuscript

to make it more complete. Based on what was said above I recommend a revise

and resubmit.

Author Response

Thank you for these suggestions. I was not aware of (Reinoso-Carvalho et al. 2020) which is a relatively recent publication. But I lean towards Bayesian methods wherever they lend themselves to a specific problem. I have included several paragraphs in the “Conclusions and Discussion” section that summarize (Reinoso-Carvalho et al. 2020)s methodology and applications. I show how these ideas can be effectively applied in the context of my current research paper to address the problems I have raised in interpreting Likert-scaled datasets.

Reviewer #2:

The paper focuses on minimizing the bias and informativeness of Likert metrics distribution

using the Kullback-Leibler divergence, KLD, from the actual responses empirical distribution 

and an assumed distribution of respondent beliefs through the information penalty.

 KLD is a divergence measure, not a distance metric. Thus, the order of the two distributions

 matters. KLD(Q, P) is not equal to KLD(P, Q). The use of divergence and distance is

 confusing. Moreover, in the setup, it is unclear how the order of the distributions is.

Author Response

Thank you for pointing this out. I have addressed this by replacing the KLD metric with a comparable Jeffreys distance metric in the current revision of my paper (with recalculations for the tables and graphs). The Jeffreys distance is a symmetrized modification of the KLD metric, 𝐽(1,2)=𝐼(1:2)+𝐼(2:1) actually introduced prior to (Kullback and Leibler 1951) paper, in (Jeffreys and Jeffreys 1946). It is a proper distance metric that satisfies the triangle inequality. I discuss, at greater length below, the revision of my calculations around Jeffreys’ distance.

The use of the term “divergence” as statistical distances are called, has varied significantly over time, and current usage was established in (Amari and Nagaoka 2000). (Kullback and Leibler 1951) actually used “divergence” to refer to the symmetrized divergence already defined and used in (Jeffreys and Jeffreys 1946), where (Jeffreys and Jeffreys 1946) referred to this as “the mean information for discrimination … per observation” while (Lindley 1959) referred to the asymmetric function as the “directed divergence.”

Although calling KLD a “distance” is not technically wrong (it is a statistical distance in the normal usage of the term) I do think there is merit in the reviewers suggestion to use a symmetric measure for this research, as there may be a question about the ordering of the distributions in calculations. I have recalculated and restated all of the graphs, tables and results in this new revision in terms of Jeffreys divergence.

I was curious myself whether there was asymmetry in KLD versus the symmetric Jeffreys divergence for my research dataset. I took my Airline Customer Satisfaction dataset and merged the 15 columns if Likert-scaled responses into a single variable, and compared these each to a 𝑁𝑜𝑟𝑚𝑎𝑙(3,1),𝑃𝑜𝑖𝑠𝑠𝑜𝑛(3) and 𝐵𝑒𝑡𝑎(.5,.5) random variable. Summaries of linear regressions of the Jeffreys distances against KLD show little difference between Jeffreys divergence and KLD for this dataset.

## 

## Call:

## lm(formula = KL_div_norm ~ j_div_norm, data = dist)

## 

## Residuals:

## Min 1Q Median 3Q Max 

## -2.026e-14 4.000e-19 3.100e-18 5.900e-18 2.519e-16 

## 

## Coefficients:

## Estimate Std. Error t value Pr(>|t|) 

## (Intercept) 8.129e-16 9.901e-18 8.211e+01 <2e-16 ***

## j_div_norm 1.000e+00 3.533e-17 2.830e+16 <2e-16 ***

## ---

## Signif. codes: 0 '***' 0.001 '**' 0.01 '*' 0.05 '.' 0.1 ' ' 1

## 

## Residual standard error: 2.577e-16 on 6186 degrees of freedom

## Multiple R-squared: 1, Adjusted R-squared: 1 

## F-statistic: 8.012e+32 on 1 and 6186 DF, p-value: < 2.2e-16

Comparing Jeffreys divergence to KLD for Airline Customer Statisfaction Dataset

Comparing Jeffreys divergence to KLD for Airline Customer Statisfaction Dataset

## 

## Call:

## lm(formula = KL_div_pois ~ j_div_pois, data = dist)

## 

## Residuals:

## Min 1Q Median 3Q Max 

## -1.121e-14 -1.600e-18 1.700e-18 5.000e-18 1.377e-16 

## 

## Coefficients:

## Estimate Std. Error t value Pr(>|t|) 

## (Intercept) 3.613e-16 4.482e-18 8.061e+01 <2e-16 ***

## j_div_pois 1.000e+00 1.479e-17 6.760e+16 <2e-16 ***

## ---

## Signif. codes: 0 '***' 0.001 '**' 0.01 '*' 0.05 '.' 0.1 ' ' 1

## 

## Residual standard error: 1.428e-16 on 6186 degrees of freedom

## Multiple R-squared: 1, Adjusted R-squared: 1 

## F-statistic: 4.57e+33 on 1 and 6186 DF, p-value: < 2.2e-16

Comparing Jeffreys divergence to KLD for Airline Customer Statisfaction Dataset

Comparing Jeffreys divergence to KLD for Airline Customer Statisfaction Dataset

## 

## Call:

## lm(formula = KL_div_beta ~ j_div_beta, data = dist)

## 

## Residuals:

## Min 1Q Median 3Q Max 

## -1.496e-16 -7.100e-18 -1.900e-18 3.900e-18 1.009e-14 

## 

## Coefficients:

## Estimate Std. Error t value Pr(>|t|) 

## (Intercept) 2.710e-16 4.211e-18 6.435e+01 <2e-16 ***

## j_div_beta 1.000e+00 8.445e-18 1.184e+17 <2e-16 ***

## ---

## Signif. codes: 0 '***' 0.001 '**' 0.01 '*' 0.05 '.' 0.1 ' ' 1

## 

## Residual standard error: 1.29e-16 on 6186 degrees of freedom

## Multiple R-squared: 1, Adjusted R-squared: 1 

## F-statistic: 1.402e+34 on 1 and 6186 DF, p-value: < 2.2e-16

Comparing Jeffreys divergence to KLD for Airline Customer Statisfaction Dataset

Comparing Jeffreys divergence to KLD for Airline Customer Statisfaction Dataset

The assumed distributions used in the paper are standard for the beliefs. However, in real

applications, the implementation of much more complex distributions is appropriate and

natural. For example, a mixture of distributions, zero-inflated distributions, among others.

Author Response

I agree with the reviewer’s assertion “… assumed distributions used in the paper are standard for the beliefs” and indeed the fact that such assumptions may be inappropriate for a given dataset is one of the main motivations of this research. Mixture and zero-inflated distributions of the observations (versus theoretical assumptions) are, indeed quite common, if not the norm in Likert-scaled datasets. They come about because of, among other things, tribalism (mixture distributions) and belief polarization (zero-inflated distributions). Though they may exist, I haven’t seen a study that interprets Likert-scaled observations with anything but a Gaussian assumption (not that these don’t exist; I just haven’t come across one). One reason for this may be that a major portion of survey research is conducted by researchers who rely (blindly perhaps) on accessible statistical software. Indeed, complex causal chains are commonly addressed via structural equation model or regression model software that rely on an underlying assumptions of Gaussian data.

Polarized responses are quite common in research on political and social issues in North America, and thus the problem addressed is an important one. They may be modeled as zero-inflated Gaussian or Poisson distributions, though too often no accommodations are made at all for the analysis for the zero-inflated data. Survey bias towards boundary-inflated responses among polled Americans, and midpoint-inflated responses among Asians have been repeatedly documented and called out as a challenge to survey based research (e.g., see (Lee et al. 2002), (Grandy 1996) and (Wang et al. 2008)) with the Pew Research Center (Gao 2016) describing such bias as a major challenge to democracy and a consistent problem in their surveys.

How should we interpret Likert-scaled data when we have strong evidence that its distribution is zero-inflated, or otherwise highly non-Gaussian? (Reinoso-Carvalho et al. 2020) have successfully applied a Bayesian multi-response version of the multivariate logit-normal regression model to interpret complex sensory data. I think (Reinoso-Carvalho et al. 2020)’s methodology offers a generalized, though more complex and computationally intensive, method for extracting relevant information from Likert datasets that are highly non-Gaussian.

Cross-sensory response research in (Reinoso-Carvalho et al. 2020), specifically studies in the human taste response to music, has pioneered Bayesian alternatives to frequentist analysis of Likert-scaled data. In (Reinoso-Carvalho et al. 2020) a sample of 1611 participants tasted one sample of chocolate while listening to a song that evoked a specific combination of cross-modal and emotional consequences. The researchers addressed difficulties in interpreting frequentist statistical tests using discrete, categorical responses by applying a Bayesian model to quantify the information content of a response. The approach used in (Reinoso-Carvalho et al. 2020) is well suited to sentiment analysis problems that have long been analyzed using structural equation models and frequentest Neyman-Pearson hypothesis tests (Westland 2019), (Sarstedt and Ringle 2020).

Data collected for (Reinoso-Carvalho et al. 2020) study showed strong non-symmetric behavior among the bounded scales, with large numbers of respondents selecting extreme values close to the boundaries, which contradicted the assumptions of traditional multivariate regression approaches to analysis, because residuals could not be Gaussian distributed were responses at the boundaries of the response space.

In order to overcome this problem (Reinoso-Carvalho et al. 2020) remapped each outcome 𝑗 for each individual 𝑖 into a unit (0,1) range. They then used Bayesian, multi-response, multivariate, logit-normal distribution with outcome-specific intercepts and slopes, and common covariance structure across outcome measures, following the methodology in (Lesaffre, Rizopoulos, and Tsonaka 2007). The logit-normal distribution can take a variety of shapes, e.g., U-shapes and J-shapes. More importantly, they are designed to specifically address the zero-inflated data distributions that arise in particularly polarized survey responses.

A Bayesian multi-response version of the multivariate logit-normal regression model was used in (Reinoso-Carvalho et al. 2020). Outcome-specific intercepts and slopes were needed since the association of each co-variate with each of the responses could significantly differ. They also take advantage of the inherent high-correlation of responses due to individual consistency in responses, and to social and cultural clustering of beliefs (and responses) in survey data, through joint modeling of all the outcomes, allowing the borrowing of information between responses.

Their Bayesian multi-response version of the multivariate logit-normal regression model provides a flexible, scalable and adaptive model where reliance on the central limit theorem can be questionable. Additionally, where available, they provide a natural form to incorporate any prior information available, either from prior studies or from expert opinion. The transformations specified in (Reinoso-Carvalho et al. 2020) result in a model error term (representing features not captured by the data) which is multi-Normal, allowing for analysis with available statistical software.

Under the complexity issue, the Hessian matrix is singular, which leads to the impossibility

of measuring the information penalty. Therefore, in this situation, the solution requires the

application of different methods.

Author Response

It is important not to confuse the theoretical and observed Fisher Information matrices. Fisher information measures the amount of information that an observable random variable carries about an unknown parameter of a distribution that models that random variable. Formally, it is the variance of the score, or the expected value of the observed information. It always exists because it is based on actual measurements. It can be derived as the Hessian of the relative entropy.

It is essential to differentiate between the theoretical and observed Fisher Information matrices. The negative Hessian evaluated at the MLE corresponds to the observed Fisher information matrix evaluated at the MLE. It is incorrect to say that the observed Fisher information can be found by inverting the (negative) Hessian. The inverse of the (negative) Hessian is an estimator of the asymptotic theoretical covariance matrix, and the square roots of the diagonal elements are estimators of the standard errors. The theoretical Fisher information matrix is based on the Fisher information metric theorem which proves that KL-divergence is directly related to the Fisher information metric, with the Fisher Information Matrix being the Hessian of Kullback–Leibler divergence (KLD) between two (ideal) distributions.

Formally, let 𝑙(𝜃) be a log-likelihood function and theoretical Fisher information matrix 𝐼(𝜃) be a symmetrical (𝑝×𝑝) matrix containing the entries 𝐼(𝜃) = - ∂2∂𝜃𝑖∂𝜃𝑗𝑙(𝜃) for 1≤𝑖,𝑗≤𝑝

The Hessian is defined as 𝐻(𝜃) = ∂2∂𝜃𝑖∂𝜃𝑗𝑙(𝜃) for 1≤𝑖,𝑗≤𝑝 and is the matrix of second derivatives of the likelihood function with respect to the parameters. It follows that if you minimize the negative log-likelihood, the returned Hessian is the equivalent of the observed Fisher information matrix whereas in the case that you maximize the log-likelihood, then the negative Hessian is the observed information matrix.

The observed Fisher information matrix is 𝐼(𝜃̂ 𝑀𝐿) the information matrix evaluated at the maximum likelihood estimates (MLE). The second derivative of the log-likelihood evaluated at the maximum likelihood estimates (MLE) is the observed Fisher information (Pawitan 2001) . This is exactly what the optimization algorithms used in this research, like optim in R, return: the Hessian evaluated at the MLE. When the negative log-likelihood is minimized, the negative Hessian is returned. The estimated standard errors of the MLE are the square roots of the diagonal elements of the inverse of the observed Fisher information matrix. That is, the square roots of the diagonal elements of the inverse of the Hessian (or the negative Hessian) are the estimated standard errors.

The inverse of the Fisher information matrix is an estimator of the asymptotic covariance matrix 𝑉𝑎𝑟(𝜃̂ 𝑀𝐿)=[𝐼(𝜃̂ 𝑀𝐿)]−1 and the standard errors are then the square roots of the diagonal elements of the covariance matrix.

The main reason to be concerned with singularities in computing Fisher Information has to do with the asymptotics – a singularity implies that the usual (√𝑛)(𝜃̂ −𝜃)−→𝐷𝑁[0,𝐼(𝜃̂ )−1] is not valid. Alternative formulations are provided in (Rotnitzky et al. 2000) and give the generalized asymptotic distributions, dependent on a parameter 𝑠 and its parity (odd/even), where 2𝑠+1 is the number of derivatives of the likelihood. (Rotnitzky et al. 2000) provides a unified theory for deriving the asymptotic distribution of the MLE and of the likelihood ratio test statistic when the information matrix has rank one less than full and the likelihood is differentiable up to a specific order. This is important since the likelihood ratio test uses the asymptotic distribution.

It is unclear how is designed the strategy to set optimally the hyper-parameters of the ideal

belief distribution.

Author Response

I was not too sure to which part of the paper the reviewer was specifically referring, but I believe it was to the illustrative examples in section 2. My intention is to show what biases and information losses can possibly be introduced into the standard methods of interpreting

Section 2 was intended to be illustrative, and the parameter settings were designed to illustrate potentially extreme situations that could occur in the data. It wasn’t really my intention to explore a parameter space, as say one might do in optimization or machine learning. Instead, the Airline database provides a real world benchmark for parameters that highlights the situations most likely to be encountered in practice.

I have made this clear in the current revision of my paper.

The empirical application is outstanding and informative, from which the minimal information

loss found is under Normal beliefs in Likert scaled surveys.

Author Response

Thank you for the kind words. I hope that this revision clears up any remaining questions the reviewers may have.

References

Amari, Shun-ichi, and Hiroshi Nagaoka. 2000. Methods of Information Geometry. Vol. 191. American Mathematical Soc.

Gao, George. 2016. “The Challenges of Polling Asian Americans.”

Grandy, Jerilee. 1996. “Differences in the Survey Responses of Asian American and White Science and Engineering Students.” ETS Research Report Series 1996 (2): i–23.

Jeffreys, H, and BS Jeffreys. 1946. “Methods of Mathematical Physics, Cambridge, 192.”

Kullback, Solomon, and Richard A Leibler. 1951. “On Information and Sufficiency.” The Annals of Mathematical Statistics 22 (1): 79–86.

Lee, Jerry W, Patricia S Jones, Yoshimitsu Mineyama, and Xinwei Esther Zhang. 2002. “Cultural Differences in Responses to a Likert Scale.” Research in Nursing & Health 25 (4): 295–306.

Lesaffre, Emmanuel, Dimitris Rizopoulos, and Roula Tsonaka. 2007. “The Logistic Transform for Bounded Outcome Scores.” Biostatistics 8 (1): 72–85.

Lindley, DV. 1959. Taylor & Francis.

Pawitan, Yudi. 2001. In All Likelihood: Statistical Modelling and Inference Using Likelihood. Oxford University Press.

Reinoso-Carvalho, Felipe, Laura H Gunn, Enrique ter Horst, and Charles Spence. 2020. “Blending Emotions and Cross-Modality in Sonic Seasoning: Towards Greater Applicability in the Design of Multisensory Food Experiences.” Foods 9 (12): 1876.

Rotnitzky, Andrea, David R Cox, Matteo Bottai, and James Robins. 2000. “Likelihood-Based Inference with Singular Information Matrix.” Bernoulli, 243–84.

Sarstedt, Marko, and Christian M Ringle. 2020. “Structural Equation Models: From Paths to Networks (Westland 2019).” Springer.

Wang, Rui, Brian Hempton, John P Dugan, and Susan R Komives. 2008. “Cultural Differences: Why Do Asians Avoid Extreme Responses?” Survey Practice 1 (3): 2913.

Westland, J Christopher. 2019. Structural Equation Models. 2nd Edition. Springer.

---

## [Decision Letter · Decision Letter 1]

12 Jul 2022

Information Loss and Bias in Likert Survey Responses

PONE-D-21-19317R1

Dear Dr. Westland,

We’re pleased to inform you that your manuscript has been judged scientifically suitable for publication and will be formally accepted for publication once it meets all outstanding technical requirements.

Kind regards,

Carlos Andres Trujillo, PhD

Academic Editor

PLOS ONE

Additional Editor Comments (optional):

Reviewers' comments:

Reviewer's Responses to Questions

**Comments to the Author**

1. If the authors have adequately addressed your comments raised in a previous round of review and you feel that this manuscript is now acceptable for publication, you may indicate that here to bypass the “Comments to the Author” section, enter your conflict of interest statement in the “Confidential to Editor” section, and submit your "Accept" recommendation.

Reviewer #1: All comments have been addressed

Reviewer #2: All comments have been addressed

2. Is the manuscript technically sound, and do the data support the conclusions?

Reviewer #1: Yes

Reviewer #2: Yes

3. Has the statistical analysis been performed appropriately and rigorously? 

Reviewer #1: Yes

Reviewer #2: Yes

4. Have the authors made all data underlying the findings in their manuscript fully available?

Reviewer #1: Yes

Reviewer #2: Yes

5. Is the manuscript presented in an intelligible fashion and written in standard English?

Reviewer #1: Yes

Reviewer #2: Yes

6. Review Comments to the Author

Reviewer #1: The current manuscript has addressed major methodological issues and made it more transparent. I have enjoyed reading this final version of the manuscript.

Reviewer #2: The article provides insight into the Likert scaled response survey analysis by implementing rigorously appropriate statistical methods. It found support for the minimal information loss under the Normal beliefs assumption. The article is well written, easy to understand, and useful for decision-making. The link to prior works helps to make the author's arguments clear. Additionally, the paper promotes new questions and procedures.

7. PLOS authors have the option to publish the peer review history of their article (what does this mean?). If published, this will include your full peer review and any attached files.

Reviewer #1: **Yes: **Enrique ter Horst

Reviewer #2: **Yes: **Hector Zarate

---

## [Editor Report · Acceptance letter]

18 Jul 2022

PONE-D-21-19317R1 

Information Loss and Bias in Likert Survey Responses 

Dear Dr. Westland:

I'm pleased to inform you that your manuscript has been deemed suitable for publication in PLOS ONE. Congratulations! Your manuscript is now with our production department. 

Kind regards, 

on behalf of

Dr. Carlos Andres Trujillo 

Academic Editor

PLOS ONE